# Exergy Analysis of Gas Switching Chemical Looping IGCC Plants

**Carlos Arnaiz del Pozo [1], Ángel Jiménez Álvaro [1,*], Jan Hendrik Cloete [2], Schalk Cloete [2] and Shahriar Amini [2]**

[1]  Universidad Politécnica de Madrid, Calle Ramiro de Maeztu 7, 28040 Madrid, Spain; cr.arnaiz@upm.es
[2]  SINTEF Industry, Richard Birkelands vei 2B, 7034 Trondheim, Norway; henri.cloete@sintef.no (J.H.C.); schalk.cloete@sintef.no (S.C.); Shahriar.Amini@sintef.no (S.A.)
*   Correspondence: a.jimenez@upm.es

**Abstract:** Integrated gasification combined cycles (IGCC) are promising power production systems from solid fuels due to their high efficiency and good environmental performance. Chemical looping combustion (CLC) is an effective route to reduce the energy penalty associated with $CO_2$ capture. This concept comprises a metal oxygen carrier circulated between a reduction reactor, where syngas is combusted, and an oxidation reactor, where $O_2$ is withdrawn from an air stream. Parallel to CLC, oxygen carriers that are capable of releasing free $O_2$ in the reduction reactor, i.e., chemical looping oxygen production (CLOP), have been developed. This offers interesting integration opportunities in IGCC plants, replacing energy demanding air separation units (ASU) with CLOP. Gas switching (GS) reactor cluster technology consists of a set of reactors operating in reduction and oxidation stages alternatively, providing an averaged constant flow rate to the gas turbine and a $CO_2$ stream readily available for purification and compression, and avoiding the transport of solids across reactors, which facilitates the scale up of this technology at pressurized conditions. In this work, exergy analyses of a gas switching combustion (GSC) IGCC plant and a GSOP–GSC IGCC plant are performed and consistently benchmarked against an unabated IGCC and a precombustion $CO_2$ capture IGCC plant. Through the exergy analysis methodology, an accurate assessment of the irreversible loss distribution in the different power plant sections from a second-law perspective is provided, and new improvement pathways to utilize the exergy contained in the GSC reduction gases outlet are identified.

**Keywords:** gas switching combustion; gas switching oxygen production; exergy; IGCC; $CO_2$ capture; efficiency

## 1. Introduction

CO$_2$ capture and storage (CCS) is expected to be a key technology in order to meet the climate change targets of global warming increase below 1.5 °C with respect to pre-industrial levels [1]. In order for CCS to become a cost-effective solution for energy decarbonization, it is essential to reduce the energy penalty associated with it. A promising solution for reducing the efficiency loss of $CO_2$ abatement, which increases the specific capital cost ($/kW) of the plant and the costs related to fuel production and transport, consists of integrated gasification combined cycles (IGCC) with inherent carbon capture by means of gas fuelled chemical looping combustion (CLC) [2]. Solid fueled CLC (using coal directly) faces numerous challenges [3]; therefore, a conventional gasification step to generate a clean syngas fuel is employed in the plants investigated in the present work. A metallic oxygen carrier is transferred between interconnected fluidized reduction and oxidation reactors and exposed to syngas and air streams respectively to achieve inherent $CO_2$ capture, obtaining two outlet streams consisting of the products of combustion, where pure $CO_2$ is obtained after water condensation,

and an $O_2$-depleted air stream is used in the power cycle. In this way, the combustion of the fuel takes place in a $N_2$-free environment instead of a conventional combustion chamber. However, slow progress in the scale up of pressurized interconnected fluidized beds has been reported [4]. To overcome this challenge, a solution involving a reactor cluster operating alternatively in reduction and oxidation stages, with the metallic carrier remaining in each reactor, has been proposed [5]. A set of gas switching (GS) valves controls the reduction and oxidation cycle lengths of each reactor. Several oxygen carriers have been tested in the lab scale and a nickel oxide (NiO) with alumina ($Al_2O_3$) support has proved to be mechanically stable and fluidize well at high air reactor temperatures [6], which are required to attain high thermal efficiencies [7].

Furthermore, chemical looping oxygen production (CLOP) has been suggested as a more efficient solution compared to a cryogenic air separation unit [8,9]. This concept employs an oxygen carrier that is capable of releasing free oxygen in the reduction stage that can be effectively used for oxy-combustion or as gasification agent. In the work [10], an integration of CLOP and CLC employing packed bed (PB) gas switching reactors in an IGCC plant is presented, revealing an efficiency increase of approximately 2.5% relative to the plant employing only CLC. A subsequent techno-economic assessment [11] showed that PB-CLC power plants clearly outperform the precombustion $CO_2$ capture IGCC benchmark, but revealed a trade-off between the benefit of adding the CLOP reactors and the added complexity and increase in cost of the gasification section. This study concluded that concepts that integrate CLOP reactors will gain potential if the performance of the carrier can be improved through maximization of the $O_2$ molar fraction in the reduction outlet. [12] shows a reduction in the energy penalty for $CO_2$ capture of 1.8% points for a precombustion $CO_2$ capture IGCC plant employing CLOP operating at atmospheric pressures instead of an air separation unit (ASU). In [13], a pressurized gas switching oxygen production (GSOP) cluster was integrated in a precombustion $CO_2$ capture IGCC with approximately 6%-points higher efficiency and capture rates above 80%.

In this work, a gas switching combustion (GSC) IGCC and GSOP–GSC IGCC plant are modeled, employing bubbling fluidized beds as gas switching reactors in both clusters, while previous studies focused on plants employing packed bed reactors [10,14]. The fluidization mode allows an easier operation (activation of material and continuous replenishment) and avoids material-related challenges (strength and reactivity). Uniform temperature can be achieved throughout the reactor length, and the good mixing properties prevent fuel dilution (to avoid deposition on the carrier), but also lead to an undesired mixing of gases when switching between the oxidation and reduction stages, reducing the $CO_2$ purity of the reduction outlet and lowering the capture ratio. The performance of the GSOP reactors operating in fluidized mode is improved with respect to the packed bed configuration, increasing the concentration difference of $O_2$ between the reactor outlet streams relative to the values obtained in [10]. The trade-off between a more efficient generation of the oxidant stream and the lower $CO_2$ capture ratio (resulting of the fluidized bed clusters) will be evaluated in the present analysis for the GSOP–GSC IGCC plant.

Besides inherent $CO_2$ capture as a means to boost power plant efficiency with respect to other $CO_2$ capture pathways, the configurations with GS technology presented in this work also incorporate hot gas clean up (HGCU) for syngas desulphurization [15]. This unit consists of interconnected fluidized beds with zinc oxide as the hydrogen sulfide adsorbent. Other sorbents are required to remove contaminants such as HCl and $NH_3$ to acceptable ppm levels [16]. This treating technology is implemented in the novel power plant concepts presented, as it will be commercially available by the time gas-switching technology is deployed, whereas the benchmark models represent current attainable IGCC performance with and without CCS.

Literature studies involving IGCC with CCS and CLC focus on thermal analysis and the $CO_2$ capture performance of the plant, with an energy efficiency penalty, specific emissions, $CO_2$ avoidance, and SPECCA (Specific Primary Energy Cost of $CO_2$ Avoided) index as key performance indicators of the proposed concepts. In the present analysis, together with the previous plant parameters, an exergy analysis is performed, introducing a rational efficiency that takes into account the fact that heat and

work are not fully interchangeable. Moreover, the exergy analysis allows the identification of efficiency losses throughout all the subsections of the plant, enabling potential optimization strategies to be more effective, providing the process engineer with insights in the synthesis of alternative process line-ups and focus on items in which the greatest exergy destruction takes place, as suggested by [17].

In [17,18], an exergy analysis for IGCC plants with precombustion $CO_2$ capture employing different syngas shift technologies and $CO_2$ removal strategies is given, while [19] provides an exergy analysis of a plant employing CLC and compares it to unabated, oxyfuel, and precombustion $CO_2$ capture power plants. An exergy analysis of IGCC plants using CLOP to provide $O_2$ for post-combustion precombustion plants was performed in [20]. In the present work, the exergy analysis methodology is employed to evaluate the performance of the novel GSC IGCC and the GSOP–GSC IGCC power plants with HGCU and benchmark them against an unabated IGCC plant and an IGCC plant with precombustion $CO_2$ capture with the same modeling baselines. The specific performance of each power plant section is determined through the exergy utilization and exergy loss contribution parameters, as defined in Section 2.3.

The plants using chemical looping considered here have been proposed recently, and more insight on how these systems generate electrical power is desirable. The exergy method is a tool that allows process engineers and researchers to understand how the foreseen efficiency gains arise, as it provides a comprehensive analysis of the different sections of the plant through identification of the flows of exergy across them. While the thermal analysis of the plant is merely based on the energy conservation, the exergy analysis determines the actual performance of each plant system relative to the maximum possible attainable, taking into account the restrictions imposed by the second law of thermodynamics. With that aim, this work also presents an auxiliary modeling tool to effectively calculate the chemical and physical exergy of mixtures appearing in the energy conversion processes of the plants.

## 2. Modeling

In this section, a brief description of the gas switching reactor modeling aspects and the assumptions taken for the stationary power plant simulations is given. A general outline of the power plant cases modeled is presented, as well as the theoretical framework for the exergy analysis that was performed.

### 2.1. Reactor Simulations

Transient reactor simulations were carried out with MATLAB using a zero-dimensional model where the fluidized bed is represented as a continuous stirred tank reactor (CSTR), assuming that chemical and thermal equilibrium are reached. This assumption is justified based on the excellent mixing of fluidized beds and the size of the industrial-scale reactors that are considered. The reader is referred to earlier work for a detailed description of the modeling methodology and reactor operating conditions for the GSC [21] and GSOP [13] reactors.

Figure 1 illustrates the behavior of the GSC and GSOP reactors in the GSOP–GSC IGCC power plant and will be discussed here briefly. In the GSC reactor (Figure 1a), the oxygen carrier (nickel oxide) is first reduced by the syngas in the reduction step (0–1 on the *x*-axis), producing $CO_2$ and steam, and then oxidized in the oxidation step (1–8 on the *x*-axis). Due to the highly exothermic nature of the oxidation reaction, a surplus of air is fed to the reactor to limit the maximum reactor temperature. For the GSC in the GSC IGCC plant (not shown here), it is possible to recycle part of the depleted air from the gas turbine to the oxidation step in the GSC. This strategy limits the temperature variation over the cycle, allowing longer cycle times with a fixed maximum temperature and thereby limiting the undesired mixing between reactor steps (seen in Figure 1a after switching the inlet streams at *x*-axis values of 0 and 1), as discussed in previous publications [5,21].

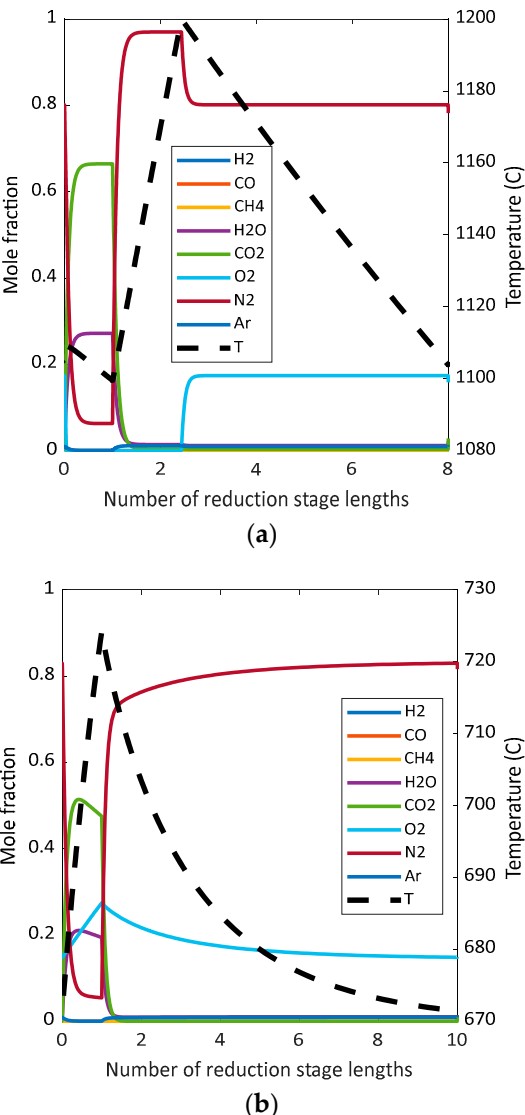

**Figure 1.** Composition and temperature over a cycle of operation for a (**a**) gas switching combustion (GSC) reactor (reduction stage length = 178.0 s) and (**b**) gas switching oxygen production (GSOP) reactor (reduction stage length = 253.7 s) in the GSOP–GSC integrated gasification combined cycles (IGCC) plant. The x-axes are normalized by the reduction stage length in each case.

In the GSOP reactors (Figure 1b), an oxygen carrier with oxygen uncoupling properties is used [10]. In the reduction step (0–1 on the *x*-axis), the oxygen carrier is reduced by syngas in exothermic reactions, heating the reactor while simultaneously releasing oxygen, which is utilized in the gasifier. In the oxidation step, part of the oxygen in the incoming air is used to re-oxidize the oxygen carrier while cooling the reactor. It can also be observed from the figure that the equilibrium oxygen concentration is dependent on the reactor temperature; therefore, the temperature variation over the cycle determined the oxygen content in the outlet streams.

*2.2. Power Plants*

A description of the models developed in Unisim Design R451 is given in this section. The benchmark cases are introduced first, and later, the GSC IGCC and GSOP–GSC IGCC plants are described. For the stationary process modeling simulations, the Peng Robinson equation of state was used to determine the thermodynamic properties of syngas/air streams, while ASME steam tables were used for the property estimations of steam in the bottoming cycle.

### 2.2.1. IGCC w/o CO$_2$ Capture

The IGCC power plant without carbon capture consists of a model similar to the one described in [22], where an elevated pressure ASU delivers oxygen to a Shell gasifier operating at the conditions described in [14] with coal loading performed with N$_2$ from the ASU. The ASU is 50% air-integrated and 100% N$_2$-integrated with the gas turbine, in order to reduce NOx emissions by lowering the combustor flame temperature. Cold gas efficiency (CGE) is approximately 80% with a carbon conversion above 99%. After HP steam generation in the syngas effluent cooler (SEC), a portion of the syngas is sent to the hot gas quench at the top of the gasifier while the remaining stream is routed to a scrubber, which eliminates the remaining particulates, ammonia, and chlorides. A COS hydrolyzer converts traces of this component to H$_2$S before entering a sulfur removal unit using absorption. Clean syngas is saturated (around 1% is extracted for coal drying), mixed with N$_2$ from the ASU and heated to 200 °C with hot water before being fed to the combustor. The gas turbine outlet is routed to a heat recovery steam generator (HRSG) with three pressure levels and intermediate reheat, where extra power output is achieved. Similar modeling assumptions to [22,23] were used to build the unabated benchmark model. Coal characteristics and compositions are detailed in Table A5 in the Appendix A. The reference F-class gas turbine described in [22] was calibrated with natural gas as fuel and integrated in the IGCC plant with syngas fuel operation at the same design nominal point, turbine inlet temperature (TIT), and pressure ratio, following a similar approach as in [14], which was referred to as "advanced gas turbine" assumptions. A schematic of this power plant main process units is given in Figure 2:

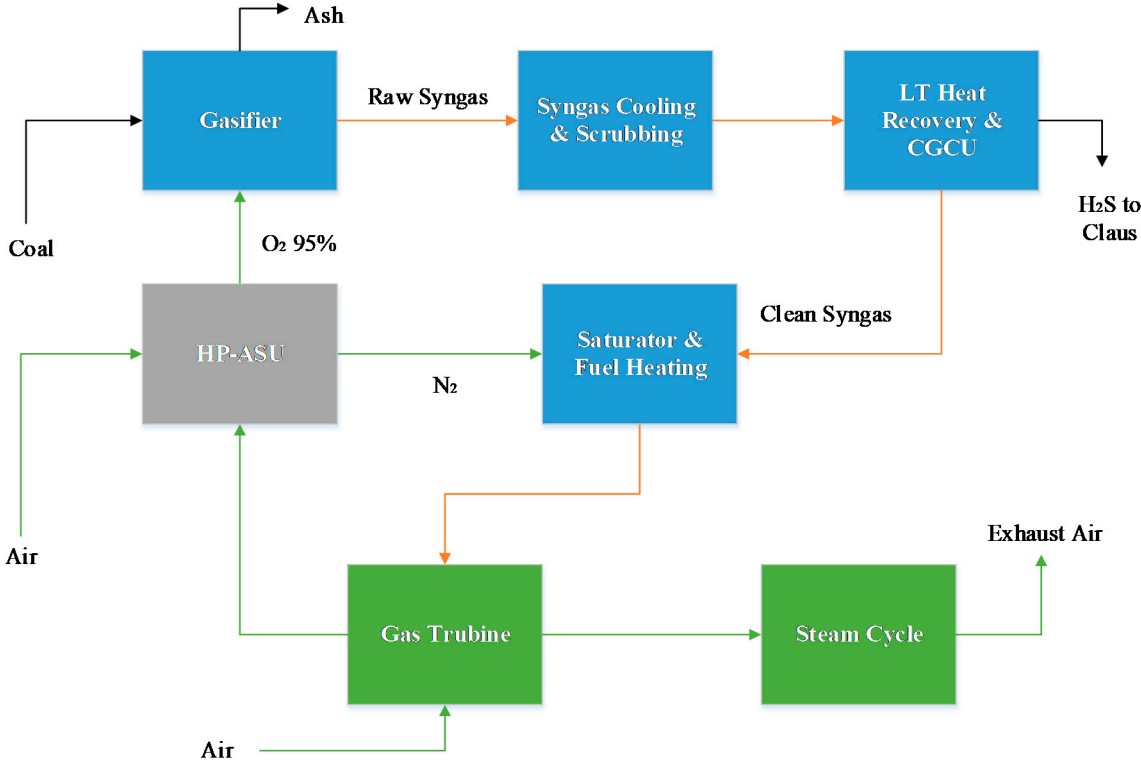

**Figure 2.** Block flow diagram of the unabated IGCC benchmark.

### 2.2.2. IGCC with Precombustion CO$_2$ Capture

In this configuration, a standalone non-integrated low pressure air separation unit (ASU) delivers oxygen (95% mol purity) to a shell gasifier as in the unabated case. Minimal or no ASU integration with the gas turbine (GT) compressor is recommended for precombustion plants [24]. Coal is loaded with CO$_2$ available from the CO$_2$ compression train. The oxygen consumption has been set equal to the unabated IGCC plant, resulting in a similar CGE. Syngas is cooled and sent to a water scrubber

to remove remaining particulate material. A water gas shift (WGS) unit consisting of two adiabatic reactors with intercooling achieves approximately 98% conversion of CO after steam addition from the high pressure (HP) stage steam turbine outlet. The steam to CO ratio was set to 1.9, and it was assumed that thermodynamic equilibrium was reached at reactor outlets. $CO_2$ and $H_2S$ are removed with a selective physical sorbent (Selexol), which was modeled with the parameters from [25], and the remaining $H_2$ is saturated with water, reheated, and diluted with $N_2$ available from the ASU before being fed to the gas turbine combustor. The TIT, pressure ratio, and blade-cooling flow fractions were assumed to be the same as that for the unabated plant. Exhaust gas heat is retrieved in an HRSG (with three pressure levels and reheating) raising steam fed to a steam turbine. The $CO_2$ absorbed in the Selexol treating unit is compressed in a five-stage intercooled $CO_2$ compressor. Similar modeling assumptions to [22,23] were used to build the precombustion $CO_2$ capture benchmark model. A schematic of the power plant is given in Figure 3.

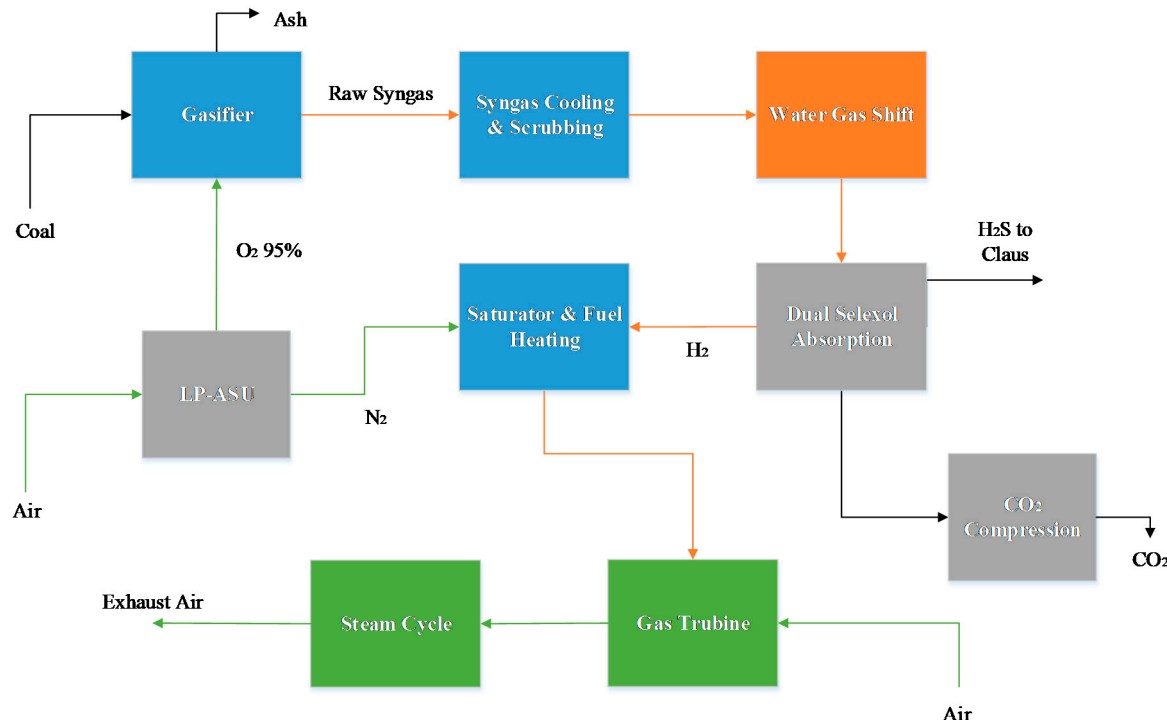

**Figure 3.** Block flow diagram of the IGCC with pre-combustion $CO_2$ capture benchmark.

### 2.2.3. GSC IGCC Plant

The GSC IGCC power plant has been presented in [5,21], and a schematic of the main components is shown in Figure 4. The gasification island is similar to the precombustion $CO_2$ capture model: a standalone ASU delivers oxygen to the gasifier, while coal is loaded to the gasifier with a fraction of the captured $CO_2$ via lock hoppers. The HGCU unit operating at 400 °C eliminates sulfur and other contaminants from the syngas; then, this stream is delivered to the GSC cluster. The sorbent regeneration in the desulfurizer unit is done by diluting an air stream with pure $N_2$ from the ASU in order to prevent sulfate formation in the zinc sorbent. The clean syngas is fed to the GSC cluster at high temperature after subtracting around 1% of the flow for coal drying. A total bed pressure of 0.75 bar was assumed. In order to maintain a high average oxidation temperature, which is critical for high thermal efficiency, whilst achieving a high carbon capture rate, the GSC cluster is operated with an $N_2$ recycle from the HRSG outlet to the gas turbine compressor after cool down in order to reduce the $O_2$ mole fraction inlet to the GSC, allowing the oxidation reaction to take place throughout the total time length of the cycle, as suggested in [26]. A slightly higher turbine back pressure (2 kPa) was specified to account for the pressure losses due to this extra cooling requirement. Afterwards,

the hot $N_2$ stream from the GSC oxidation outlet is expanded in the gas turbine, operating at the same pressure ratio as the reference plants. The HSRG unit raises steam, which is superheated in the hot reduction gases heat recovery unit, since the attainable TIT in the GSC oxidation is below 1200 °C (imposed by the oxygen carrier degradation at higher temperatures); therefore, the exhaust gases upon expansion in the GT are at a relatively low temperature (~500 °C). The cooling flow model for the stator of the GSC IGCC plant was calculated with a correlation from [27] and, due to the relatively low TIT, it was assumed that no rotor cooling is required. The GT component polytropic efficiencies are the same as the ones employed in the benchmark plants to allow a consistent comparison.

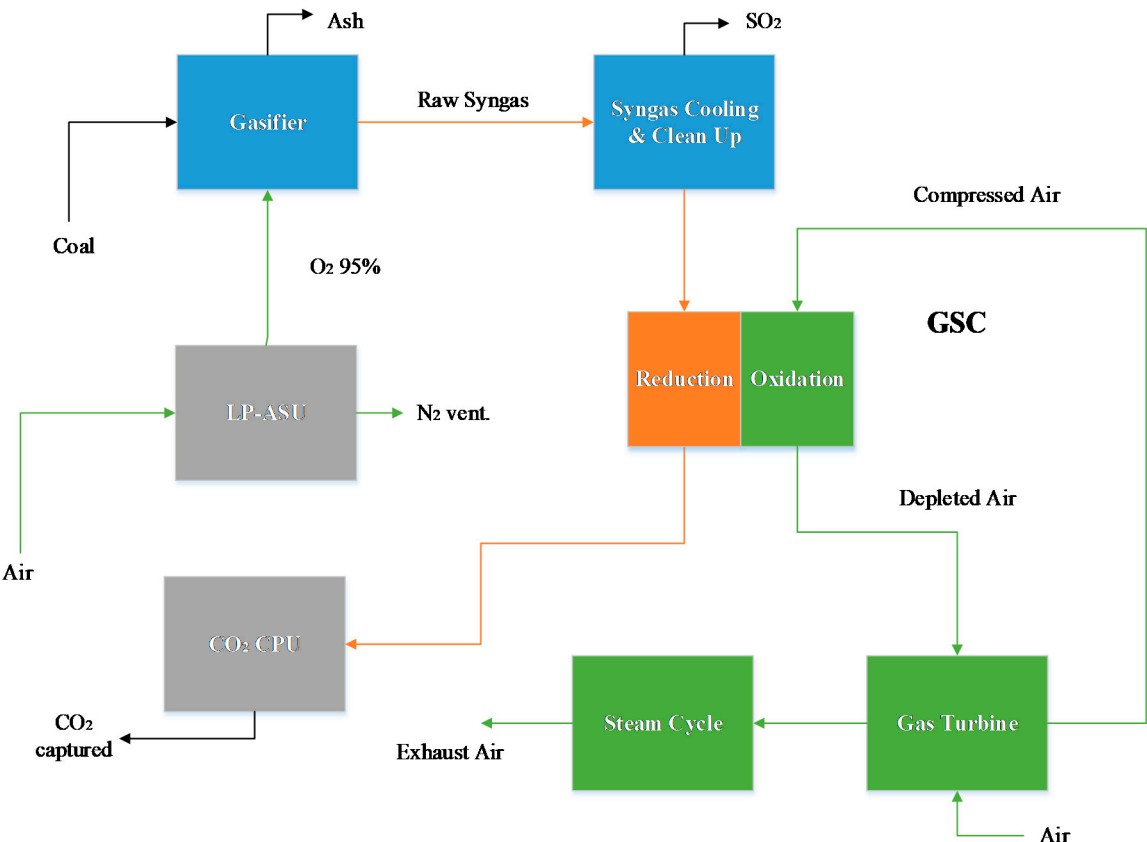

**Figure 4.** Block flow diagram of the GSC IGCC plant.

The reduction gases outlet is further cooled down in a condenser and water is knocked out, prior to a $CO_2$ purification unit ($CO_2$ CPU) consisting of a two-flash vessel cryogenic system similar to the one described in [28,29]. This purification step is needed because, due to the mixing occurring during the valve switch in the GSC reactors, the $CO_2$ purity resulting after water removal is below the 96% mol. specification required for $CO_2$ capture and storage [22,30]. A supercritical $CO_2$ pump delivers a sequestered $CO_2$ stream at 150 bar.

### 2.2.4. GSOP–GSC IGCC

The IGCC power plant employing a GSOP reactor cluster instead of an ASU and GSC in place of a combustion chamber was modeled taking into account the following considerations.

The purity of the oxidant stream (~23%mol $O_2$) is low compared to that delivered by an ASU; therefore, the gasification technology employed consists of a circulating fluidized bed Winkler gasifier [31] operating at substantially lower temperature than the entrained flow Shell gasifier technology of the benchmark and GSC IGCC power plants. If an entrained flow gasifier was coupled to a GSOP reactor cluster, a very low CGE would be obtained, as a substantial heating value of coal would be invested in heating the species in the gasifier to high temperatures (>1400 °C). The fluidized

bed gasifier was modeled assuming a syngas outlet temperature of 900 °C and fixing the ratio of the lower heating value (LHV) of methane in the syngas and coal LHV input to 11.3%, analogously to [10]. The carbon conversion achieved in the Winkler gasifier is lower than in the Shell gasifier process, and a value of 95.5% was used. Since the Winkler gasifier operates at a slightly lower pressure than the GT pressure ratio, the oxidizing stream delivered by the GSOP at elevated temperature can be effectively coupled with the gasification system (allowing for a slight oxidizer overpressure of 0.5 bar). Consequently, a booster compressor operating at a high inlet temperature must be introduced before the GS cluster to overcome the pressure losses of the gasifier–GSOP loop. The pressure drop assumed in each reactor cluster was consistent with the GSC IGCC case.

The GSOP–GSC IGCC was also modeled with HGCU for sulfur and contaminant removal operating at 400 °C, but in this case, the reduction of the $O_2$ mole fraction in the regeneration stream is accomplished by partially recirculating part of wet flue gas desulfurization (WFGD) outlet to the regeneration stream compressor inlet, as suggested in [10]. The fraction of syngas delivered to the GSOP reactor was adjusted to achieve the specified reduction outlet temperature. The total air flow rate was manipulated to set the GSC oxidation temperature to the fixed value of 1150 °C, which gives a reasonable trade-off between thermal efficiency and $CO_2$ capture. The portion of the recirculated GSC reduction outlet used as sweep gas in the GSOP was varied in order to achieve the desired gasifier outlet temperature and specified heat loss. The model reached the simultaneous convergence of GSOP, GSC, and gasifier blocks, meeting all design/operation specifications with error tolerances below ~0.5 °C. The oxygen-depleted air stream is expanded in the gas turbine (GT), and subsequently, heat is recovered in an HRSG unit for power generation in the steam turbine. Analogously to the GSC IGCC plant, most of the steam superheating is carried out with the hot reduction gases heat recovery unit, which allows steam temperatures to reach 565 °C. In addition, the stator cooling correlation assuming a blade temperature of 850 °C was employed, neglecting the rotor cooling flows due to the low operating temperatures. The reduction gases outlet after heat recovery and water knock out is sent to the $CO_2$ CPU to meet the transport and storage specifications. A schematic of the GSOP-GSC plant is given in Figure 5:

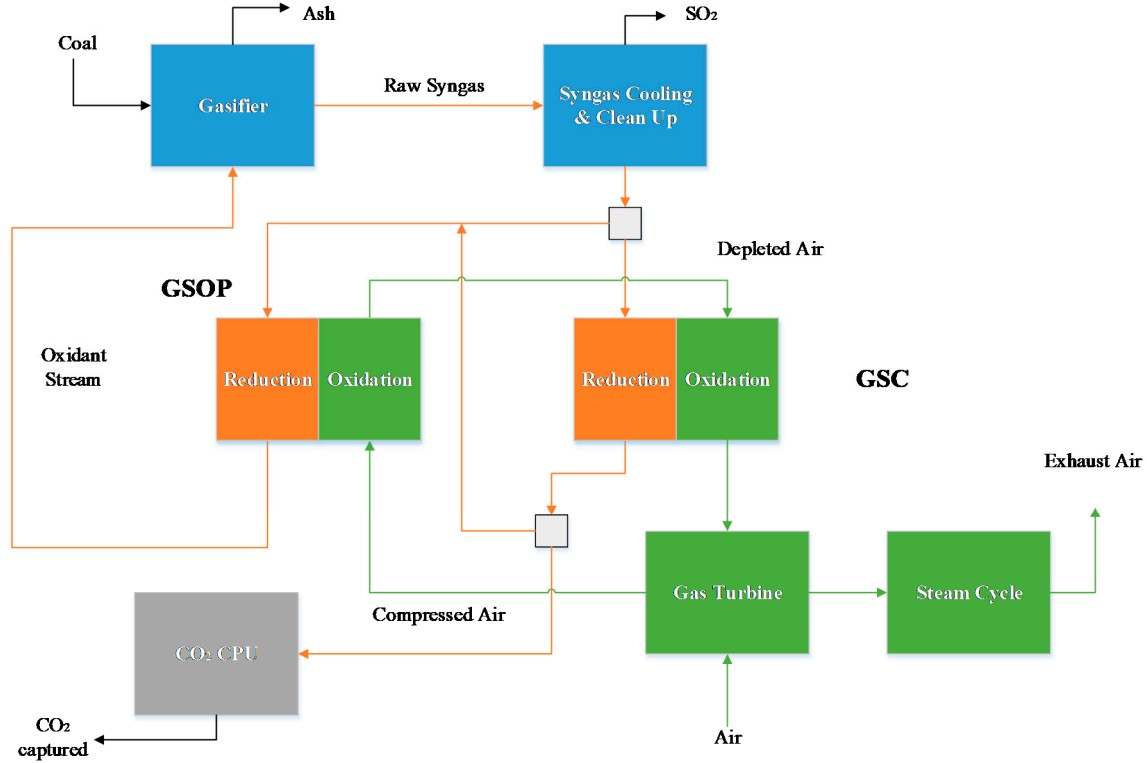

**Figure 5.** Block flow diagram of the GSOP–GSC IGCC power plant.

In all the models, a fixed coal input is assumed, resulting in net plant power outputs between 300 and 450 MW. This approach is consistent with other studies involving the exergy analysis of plants [17,19]. Further modeling assumptions of the power plant can be found in the Appendix A.

### 2.3. Exergy Analysis Tool

The exergy balance suggested in [32] for each system *j* of the power plant is given by Equation (1) under stationary conditions and assuming that the changes in kinetic and potential energy are negligible. A system may comprise one or more unit operations defined by a control volume, which sets the boundary to which Equation (1) is applied.

$$0 = \dot{Q}_j\left(1 - \frac{T_0}{T}\right) - \dot{W}_j - \dot{I}_j + \sum_i \dot{E}_{i,j}$$ (1)

where $T_0$ and $P_0$ are the temperature and pressure at ambient conditions. The total exergy flow will is calculated as $\dot{E}_{i,j} = e_{i,j}\dot{m}_{i,j}$. The exergy destruction term can be determined when the work $\dot{W}_j$ and heat flows $\dot{Q}_j$ alongside the specific exergy flow for stream *i* entering or leaving system *j*, $e_{i,j}$ are known. The exergy flow terms are calculated with Equation (2).

$$e_i = e_{Ch} + e_{Ph}$$ (2)

$$e_{ph} = h_i - T_0 s_i - (h_0 - T_0 s_0)$$

The determination of the chemical and physical exergy ($e_{Ch}$, $e_{Ph}$) employing the simulation software requires the creation of an auxiliary tool which accounts for the following:

1.  A series of mechanical transformations that bring a given stream mixture at certain pressure $P_i$ and temperature $T_i$ to the reference environment values ($P_0$ and $T_0$), so that the physical exergy term can be determined by accessing specific enthalpies and entropies: $h_i, s_i$.
2.  A series of mechanical and chemical reaction elements that withdraw co-reactants from the reference environment, react with the mixture at $T_0$ and $P_0$, and release the products of the reaction to the environment at the same chemical potential (i.e., partial pressure) at which these components are found in the environment. If no devaluation reactions take place in the components of the mixture, the tool determines the chemical exergy as the concentration imbalance of the components of the mixture with respect to the environment composition. Figure 6 shows a simplified block diagram of the rationale for the determination of the chemical and physical exergy of a mixture:

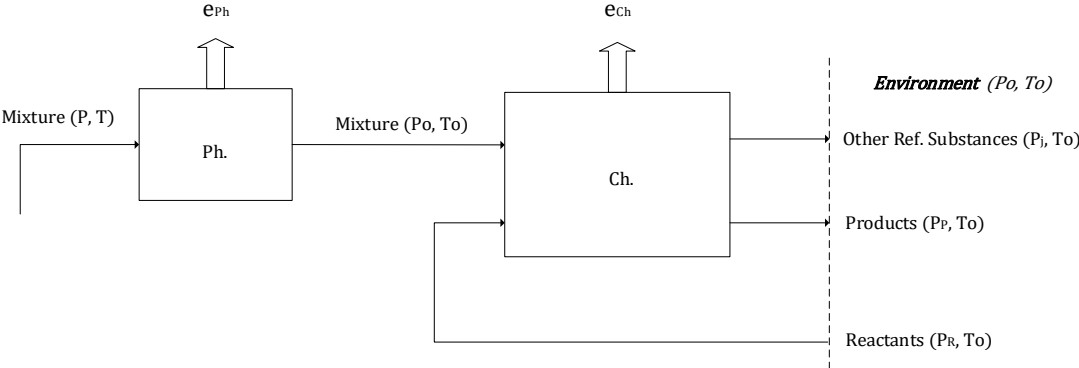

**Figure 6.** Schematic of the modeling tool employed to determine the chemical exergy of a stream.

When heat is rejected to the environment across the boundary of the power plant, such as in a steam condenser, it is assumed that the heat term in Equation (1) is released to a reservoir of infinite

mass permanently at ambient conditions. Therefore, any heat stream leaving the power plant is considered an exergy destruction inherent to the plant subsystem stream source, as no useful work is retrieved from it. To illustrate this, if we consider the steam condenser of the actual plant, the cooling water will experience an exergy gain due to a small temperature rise. However, this exergy is later destroyed/lost in the cooling water tower auxiliary unit, and it is not practical to implement a thermal machine from which power can be produced from that stream.

The environment composition is defined with a dry air composition, vapor pressure correlation for water [33], and the relative humidity. A 60% relative humidity was assumed in the calculations. The inert thermomechanical state was taken as 25 °C and 1.01325 bar. The devaluation reactions considered in the tool include all the combustion reactions of components present in syngas and natural gas. The specific properties of the streams are determined by setting a specified thermodynamic property package; in this case, it is the Peng–Robinson equation of state. The exergy tool has been validated with data from [34] and for chemical exergies of single components with errors below 2%. Reference environment composition can be found in Table A4 in Appendix A. The exergy of coal and slag leaving the gasification system was accounted for by using a correlation provided by [34], it was assumed that the chemical exergy of the ash components was negligible compared to the organic matter.

$$e^0{}_{solid} = (LHV + 2442w)\varphi_{dry} + 9417s \tag{3}$$

$$\varphi_{dry} = 1.044 + \frac{h}{c}0.1882 + \frac{o}{c}0.0610 + \frac{n}{c}0.0404$$

where *w, s, h, c, o* and *n* are the coal mass fraction of water, elemental sulfur, hydrogen, carbon, oxygen, and nitrogen, respectively.

The performance of each individual plant block can be expressed by means of the exergy utilization parameter $\lambda$, which is defined as the ratio between the sum of all exergy output from the block relative to the exergy input. It gives an idea of the degree of thermodynamic reversibility of the processes taking place in that block, and it must be evaluated together with the relative weight of the exergy destruction of that block with respect to the total of the plant, or exergy destruction contribution $\delta$, for a comprehensive understanding.

$$\lambda_j = \frac{\dot{W}_{j,out} + \sum_{out} \dot{E}_{i,j}}{\left|\dot{W}_{j,in}\right| + \sum_{in} \dot{E}_{i,j}} \tag{4}$$

$$\delta_j = \frac{\dot{I}_j}{\sum \dot{I}_j} \tag{5}$$

The overall exergy efficiency is defined as the ratio of the useful effect and the exergy input to the plant (provided by the coal). As a useful effect, the net power output is usually considered. However, for the plants with CCS, the exergy leaving the plant as high-purity $CO_2$ can be considered useful, if such a stream could later be employed in a subsequent chemical plant (e.g., through the Sabatier reaction), redefining the exergy efficiency as follows:

$$\xi = \frac{\dot{W}_{net}}{\dot{E}_{coal}} = 1 - \frac{\sum \dot{I}_j + \sum \dot{E}_{out}}{\dot{E}_{coal}} \qquad\qquad \xi' = \frac{\dot{W}_{net} + \dot{E}_{CO2}}{\dot{E}_{coal}} \tag{6}$$

## 3. Results

In this section, the results for the power plant simulations are given. A thermal analysis, power breakdown, and emission performance is shown, and subsequently, the exergy analysis results for the different power plants are detailed.

*3.1. Plant Power Breakdown and CO₂ Emissions Performance*

Table 1 shows the results of the models for the different power plants considered. It can be seen that the energy penalty associated with $CO_2$ capture is approximately 10% points for the precombustion $CO_2$ capture plant. The low thermal efficiency results in an elevated SPECCA index and a $CO_2$ avoided approximately 3% points below the capture rate of the plant. The benchmark plants show similar performances to [22].

The GSC IGCC plant presents an energy penalty of approximately 4% points and a high capture rate. $CO_2$ emissions (lock hopper venting, coal drying, GSC mixing, and CPU vent) result in a capture rate comparable to the precombustion plant, but with a higher $CO_2$ avoided and lower SPECCA index due to the higher thermal efficiency. It is noteworthy to mention the lower auxiliary consumption for this plant, as no $N_2$ compression is required to avoid $NO_x$ emissions. The flameless combustion in CLC has the intrinsic advantage of eliminating entirely this hazardous contaminant from these plants. Furthermore, the $CO_2$ compression duty is reduced relative to the precombustion $CO_2$ capture model, since the reduction gases are pressurized, requiring only two compression stages (and pump) to reach the delivery pressure. The efficiency results are comparable to the values obtained in [21].

For the GSOP–GSC IGCC plant, the energy penalty is minimal: around 1.3% points. The high electrical efficiency yields a $CO_2$ avoidance similar to the actual capture rate of the plant (note that the carbon conversion for the Winkler gasifier is lower than the Shell gasifier used in the unabated IGCC benchmark, leading to less $CO_2$ production), and yields SPECCA indexes below 0.5 MJ/kg of $CO_2$ captured. The energy penalty imposed by air separation units and $N_2$ compression is very pronounced in the benchmark models (they constitute the major auxiliary consumption of these plants, approximately 80% for the unabated case). In the GSOP–GSC plant, these items are avoided and, in combination with the lower $CO_2$ compression duty, the total auxiliary consumption is approximately 30% of that of the precombustion $CO_2$ capture plant. The combination of the lower TIT reached with GSC cluster and the fact that the reduction gases' GSC outlet is not expanded, (some of the heat is effectively recovered as useful work by raising steam in the hot gases heat recovery unit), results in a larger proportion of the power output being delivered by the steam cycle relative to the gas turbine compared to the reference plants. The precombustion $CO_2$ capture model shows the lowest steam power output, as a substantial amount of IP steam is extracted and delivered to the WGS unit. As shown in Table 1, the lower TIT of the GS plants results in a higher flow rate across the turbomachinery elements for the same total heat input, while the exhaust flow is reduced due to the subtraction of $O_2$ in the GS clusters.

Finally, when comparing the GSOP–GSC IGCC plant to the results shown in [10], where packed bed reactors are used, several aspects can be noted. The greater $O_2$ concentration difference achieved in the GSOP reactors results in an oxidizing stream with a higher oxygen purity (23.0% mol compared to 16.8% mol), and the resulting syngas has a smaller flow rate and higher heating value comparatively. As a consequence, the heating value of the syngas can be more effectively employed in heating a higher mass flow rate of air through the gas turbine, while less HP steam is generated in the SEC. A slight efficiency improvement and a shift of power output toward the topping cycle is observed. Additionally, the different GSOP cluster operation alters the relative fractions of syngas to the GS reactors. In this study, for the HGCU case, approximately 32% of syngas was routed to the GSOP, while the reduction gases' recirculated fraction was around 19%. Overall, the syngas mass flow rate was 20% lower than [10], with a 21% higher LHV. The $CO_2$ capture ratio is to some extent reduced by the fluidized bed operation of the GS clusters. Since the gasifier, GSOP, and GSC cluster are highly integrated, it is not possible to resort to a $N_2$ recycle from the HSRG outlet as in the GSC IGCC plant, and consequently, the high capture rate and high oxidation outlet temperatures that lead to high efficiencies cannot be achieved simultaneously. Alternative strategies such as purging with steam have been proposed to reduce the degree of mixing, reaching capture rates above 90% while sacrificing power output from the bottoming cycle, as described in [5].

**Table 1.** Power plant results. ASU: air separation unit, CPU: $CO_2$ purification unit, GT: gas turbine, SPECCA: Specific Primary Energy Cost of $CO_2$ Avoided.

| Item/Plant | Unabated IGCC | Precombustion $CO_2$ Capture IGCC | GSC IGCC | GSOP–GSC IGCC |
|---|---|---|---|---|
| Coal Input (MW) | 854.0 | 854.0 | 854.0 | 854.0 |
| GT Net (MW) | 277.8 | 271.7 | 201.3 | 203.5 |
| Air Expander | 5.5 | 0.0 | 0.0 | 0.0 |
| Steam Turbine Net (MW) | 189.3 | 157.8 | 234.0 | 222.7 |
| ASU (MW) | 26.3 | 36.8 | 36.8 | 0.0 |
| $N_2$ Compression (MW) | 28.1 | 23.7 | 0.0 | 0.0 |
| Pumps (MW) | 2.7 | 3.1 | 3.2 | 2.9 |
| Heat Rejection (MW) | 2.1 | 2.3 | 3.9 | 3.0 |
| $CO_2$ CPU/Compression (MW) | 0.0 | 23.4 | 14.6 | 11.4 |
| Coal Milling (MW) | 1.7 | 1.7 | 1.7 | 1.4 |
| Ash Handling (MW) | 0.5 | 0.5 | 0.5 | 1.1 |
| GT Aux. (MW) | 1.0 | 0.9 | 0.7 | 0.7 |
| Syngas Compressor (MW) | 1.1 | 1.2 | 1.0 | 8.1 |
| AGRU Aux. (MW) | 1.2 | 11.7 | 1.3 | 1.2 |
| Balance of Plant (MW) | 1.3 | 1.3 | 1.3 | 1.3 |
| Total Aux. (MW) | 66.0 | 104.1 | 64.8 | 31.0 |
| Gross Plant (MW) | 472.7 | 429.5 | 435.3 | 426.2 |
| Net Plant (MW) | 406.7 | 322.7 | 370.5 | 395.1 |
| Gross Efficiency (LHV %) | 55.4 | 50.3 | 51.0 | 49.9 |
| Net Efficiency (LHV %) | 47.6 | 37.8 | 43.4 | 46.3 |
| Specific Emissions ($kgCO_2$/MWh) | 727.3 | 86.4 | 62.6 | 116.4 |
| Capture Rate (%) | 0.0 | 90.6 | 92.2 | 83.9 |
| $CO_2$ Avoided (%) | 0.0 | 88.1 | 91.3 | 84.0 |
| SPECCA (MJ/$kgCO_2$) | - | 3.07 | 1.09 | 0.36 |
| Compressor Intake (kg/s) | 514.8 | 443.4 | 735.9 | 818.7 |
| Exhaust Gases (kg/s) | 605.2 | 519.4 | 690.5 | 748.7 |

## 3.2. Power Plant Exergy Analysis

Employing the analysis procedure described in the second section, the different power plant losses were identified for each of the models employing GS technology and the benchmarks. The total exergy breakdown (in MW) is given in Figure 7, showing the electrical power output, the total exergy destruction of each power plant block, and the exergy flows leaving the plant. The "Exergy Out" block includes the main exergy flows that leave the power plant: slag from the gasifier, the exhaust gases from the HSRG outlet, the $CO_2$ stream for the plants with CCS, and a group of different minor outlet streams such as water, $H_2S$ in the precombustion capture plant, vented $N_2$ from the ASU, etc. The exergy utilization and exergy destruction contribution in percent is subsequently given in Figures 8 and 9 for the benchmark and GS plants, respectively. The power plant losses are grouped in representative blocks of the different sections of the plant. The gasification island includes the gasification unit (with gas quench) and the ASU (if present). The syngas cooling and treating block include all the unit operations from syngas effluent coolers, candle filters, and syngas scrubbing, low-temperature heat recovery and desulfurization, the HGCU for the gas switching chemical looping plants, syngas boosters, fuel saturator and heating, and in the case of the precombustion model, the WGS unit. The gas turbine includes the compressor, combustion chamber, and turbine. For the plants with GS technology, lacking a combustion chamber or ASU, the GS clusters are shown separately. The steam cycle elements include the main HSRG, the condenser, the deaerator feedwater tank and pumps, the steam turbine, and the hot reduction gas recovery unit for the GS plants. The $CO_2$ compression and purification unit comprises the reduction gases condenser for water knock out, the two flash vessel purification units for the GS plants, and the $CO_2$ pump. For the precombustion $CO_2$ capture model, the five-stage intercooled compressor is accounted for here. The detailed summary of the different process units

considered in each power plant block with their respective exergy losses (total and percentage) is provided in Table A6 in Appendix A.

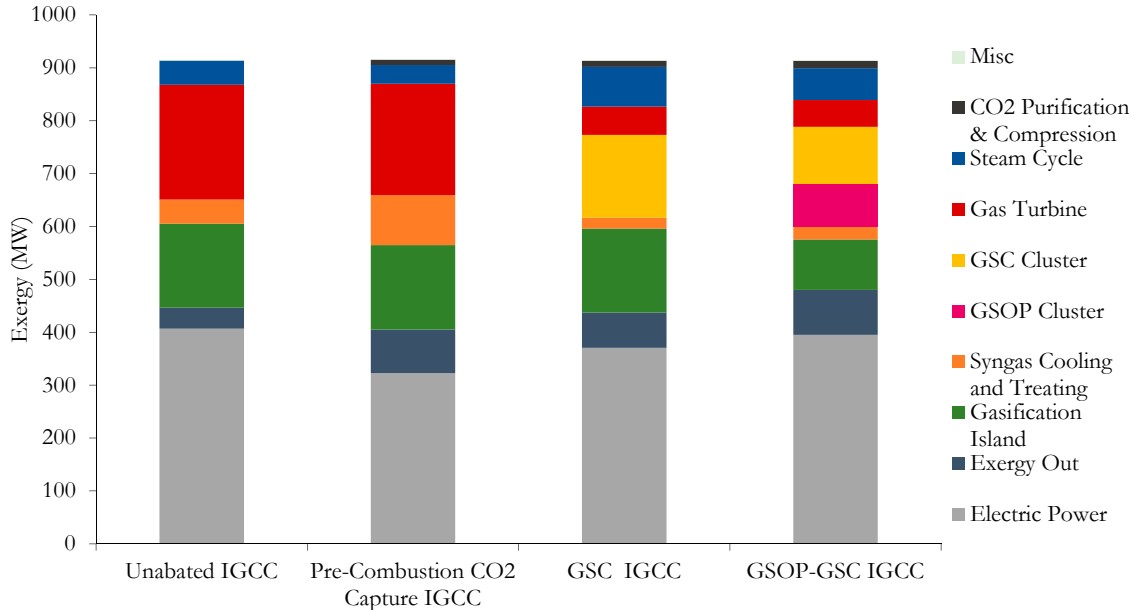

**Figure 7.** Total exergy breakdown of the different power plants.

Taking a close look at the benchmark models' exergy breakdown, it can be seen that most of the exergy destruction occurs in the gasifier and gas turbine (combustion chamber primarily), where the chemical degradation of the fuel takes place. Similar losses are found between the two models due to the similar assumptions made for these plant items. The high-exergy utilization of the gasifier island (>90%) is due to the large amount of incoming exergy to this plant section, although the exergy loss contribution surpasses 30%. The gas turbine presents a low-exergy utilization (around 70%) with a high overall loss contribution (approximately 45%). With regard to the precombustion $CO_2$ capture IGCC model, substantial losses are incurred upon by adding the shift reaction step and the acid gas removal unit (AGRU), increasing the loss contribution of this block by around 10% points, and duplicating the exergy losses in this section relative to the unabated IGCC plant. A large portion of the steam is supplied to the WGS unit; therefore, the steam cycle losses are somewhat lower comparatively. Since the precombustion $CO_2$ capture IGCC plant delivers a pure $CO_2$ stream at 150 bar, it can be seen in Figure 7 that the exergy flow leaving the system is twice the amount of the unabated IGCC model, where the only source of exergy flow leaving the plant are the exhaust gases from the HSRG and to a lesser extent the slag and captured sulfur in the absorption treating unit.

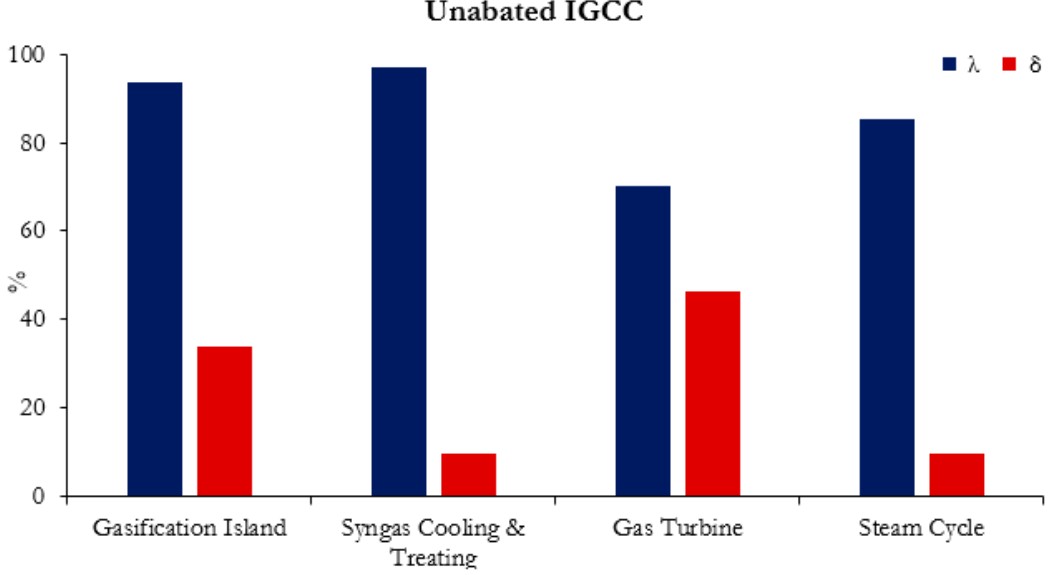

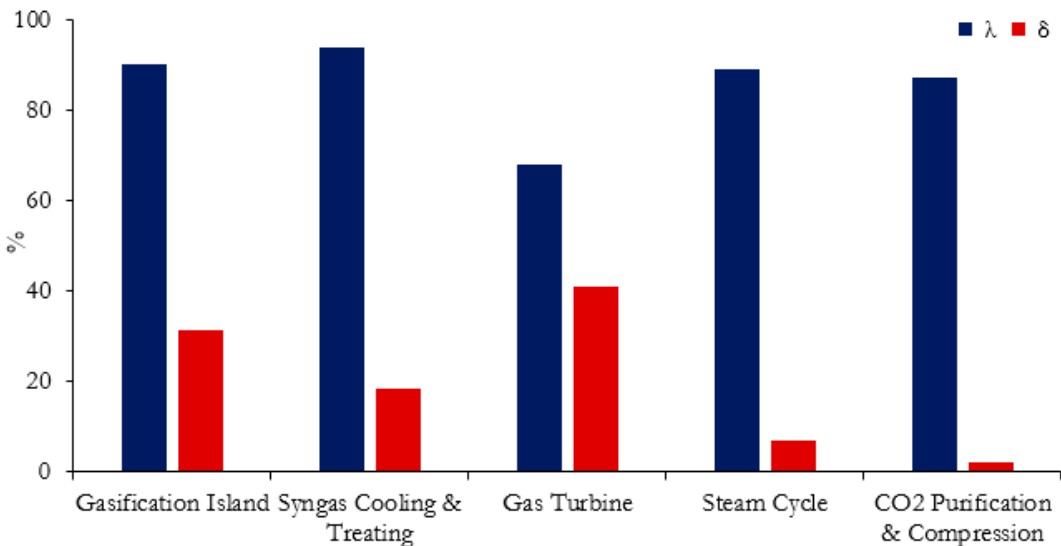

**Figure 8.** Exergy utilization λ and exergy destruction contribution δ in percent for the benchmark power plants, as defined by Equations (4) and (5).

On the other hand, for the GSC IGCC plant, it can be observed that the losses due to the GT are greatly reduced as the combustion chamber is removed, and the exergy utilization of this block significantly increases by more than 20% points. The turbomachinery elements show a different distribution of losses due to the different TIT with respect to the reference plant. The combustor losses are taken up to a great extent by the GSC cluster, with an exergy utilization and loss contribution of approximately 84% and 33%, respectively. While the gasification island remains unchanged, the effect of employing an HGCU system reduces the losses of the syngas cooling and treating section by half, relative to the unabated IGCC model. As the power ratio in the GS chemical looping plants shifts toward the steam cycle, it is observed that this item takes a greater portion of the overall exergy destruction. The extra heat recovery unit for the GSC reduction outlet and the $N_2$ cooler also contribute significantly to this increase, reducing the exergy utilization and increasing the loss contribution relative to the unabated benchmark plants by around 3% points and 5% points, respectively. Since the exhaust gases are cooled down and recycled to the GT compressor, and all the exergy in the sulfur is assumed to be lost (as full $H_2S$ combustion takes place during absorption and regeneration), the fraction of exergy

leaving the plant is reduced compared to the precombustion $CO_2$ capture model, given that both have similar $CO_2$ exergy output. In addition, the condensation of the reduction gases for the GSC-IGCC plant slightly increases the losses due to $CO_2$ purification and compression, compensating for the lower compression requirements of the $CO_2$ CPU compared to intercooled compressor of the precombustion $CO_2$ capture model, and resulting in around 7% points higher exergy utilization, but overall with a slightly higher exergy loss contribution of around 1% point.

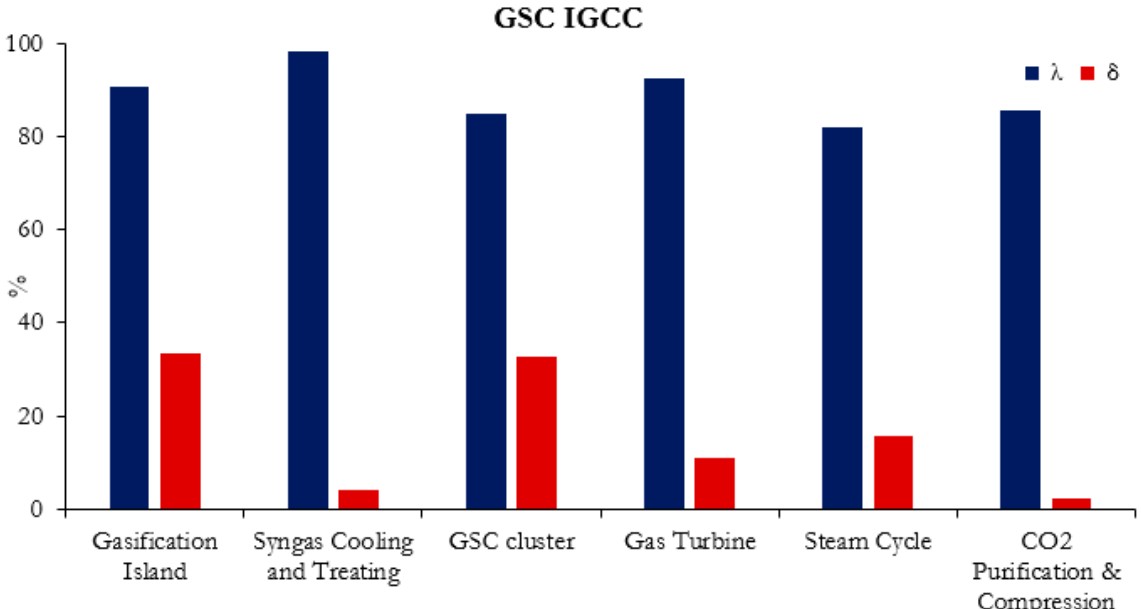

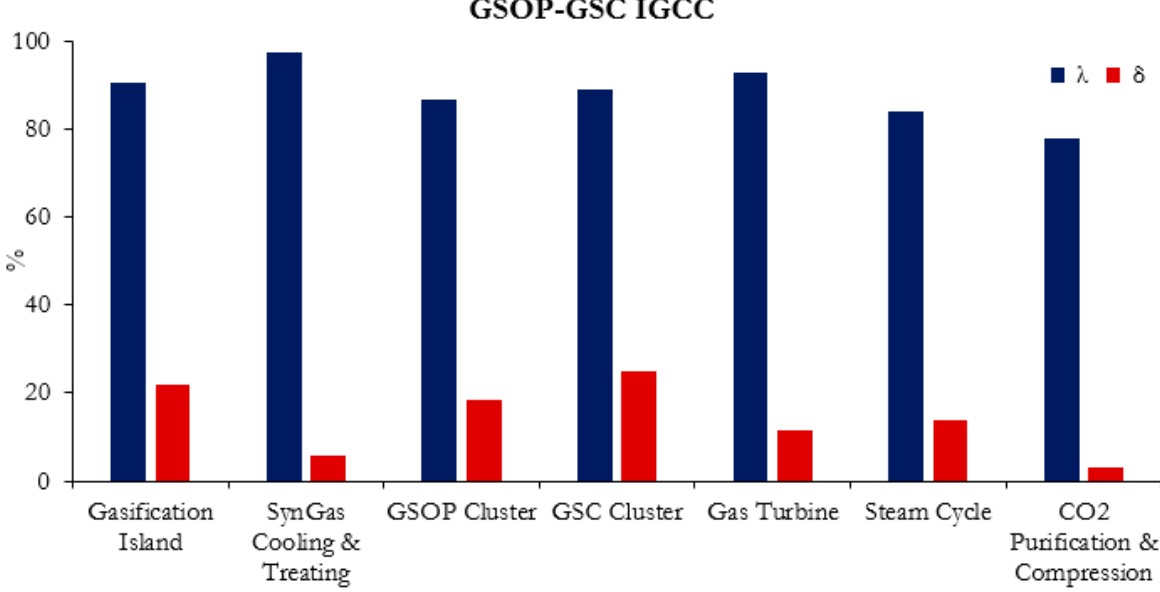

**Figure 9.** Exergy utilization $\lambda$ and exergy destruction contribution $\delta$ in percent for the gas switching (GS) power plants, as defined by Equations (4) and (5).

When analyzing the GSOP–GSC plant and comparing it to the GSC IGCC model, a great reduction in the gasifier island total exergy destruction is observed. Although the exergy utilization is similar, a reduction in the exergy loss contribution of more than 10% points is achieved. This is due to several factors: the most obvious one is the removal of the ASU with a high auxiliary consumption. Secondly, the gasification taking place with a hot oxidant stream delivered by the GSOP and occurring at lower

temperature relative to the entrained flow shell gasification plants results in a better overall conversion of coal into syngas, reducing the amount of heating value in the original fuel, which is downgraded into steam. The syngas cooling and treating losses are similar to the GSC IGCC plant: although a bigger syngas booster is required to overcome the pressure losses in the Winkler gasifier–GSOP loop, in the GSC IGCC plant, a high irreversibility occurs during the pressure letdown of the syngas from the shell gasifier to GSC pressure, which suggests that a syngas expander after HGCU can be an effective way to minimize these losses. The losses that are caused by the conversion of syngas are now distributed between GSOP and GSC clusters, which increase the temperature of the compressed air sequentially with the heat released in the oxidation stage, resulting in a similar exergy utilization as in the cluster of the GSC IGCC plant. An overall greater exergy loss between both clusters relative to the GSC IGCC plant is observed, since the recirculation of some reduction gases from the GSC reduction outlet to the GSOP results in a mixing of streams at significantly different temperatures. The GT shows a minor improvement as the compressor inlet is at a lower temperature than the recirculated (cooled) gases in the GSC IGCC model. The steam cycle has a lower exergy loss contribution, since the stack must not be further cooled and recycled; therefore, a higher portion of exergy leaves the system as hot HSRG exhaust. Although the capture rate is approximately 10% points lower, the exergy leaving the system is substantially higher. This is not only due to the aforementioned stack losses but primarily because of the larger exergy content of the slag leaving the gasification system. Notably, increasing the carbon conversion of the Winkler gasifier would lead to interesting efficiency benefits, as this exergy stream corresponds to around 3% of the total.

A relevant result observed from this study is that more exergy is destroyed in the hot reduction gases heat recovery unit than in the main HRSG (although the flow in the former is approximately one-sixth of the latter). These losses are due to the high temperature difference between the steam that is generated and hot reduction gases, despite the lower overall heat transferred compared to the HSRG. Alternative process line-ups could be conceived based on this insight. For instance, operation of the GT at a lower pressure ratio (to achieve a sufficiently high turbine outlet temperature (TOT) for steam superheating in the main HSRG) and syngas preheating with reduction gases outlet could be a possibility to be explored in future works. [21] shows the benefit of adding a recuperator between the hot gases and compressed air for a GSC IGCC plant with extra firing. Alternatively, a reduction gases expander could be employed, although a higher power demand and number of recompression stages for $CO_2$ capture will lessen the attractiveness of this option. If the reduction gases have an increased flow due to a larger presence of water, which is condensed before $CO_2$ compression (for instance in a configuration where syngas cooling is accomplished with a water quench), then the expansion of the reduction gases expander will likely be interesting, and the greater power output will compensate for the increased compression stages of the $CO_2$. The improved benefits of a hot oxidant stream for gasification are manifest from the GSC-GSOP IGCC model, which suggests that the exergy contained in the reduction gases outlet of the GSC could be effectively invested in a coal pregasification step to enhance the CGE. Preheating to 300 °C of a coal–water slurry feed in an entrained flow gasifier showed substantial efficiency improvement [35]. These alternative configurations are left for future work.

The overall exergy efficiency, based on the definitions presented in Equation (6), is compared in Figure 10, when the exergy-contained $CO_2$ captured stream is considered as a useful effect, and when only the net power output is assumed. Since the chemical exergy of fuels predicted by Equation (3) is typically above the lower heating value, this latter exergy efficiency is around 3% points below the respective thermal efficiencies given in Table 1. Interestingly, considering the former exergy efficiency definition, the GSC IGCC and the GSOP–GSC plants with $CO_2$ capture can outperform the unabated IGCC benchmark by around 1.5% and 3.5% points, respectively.

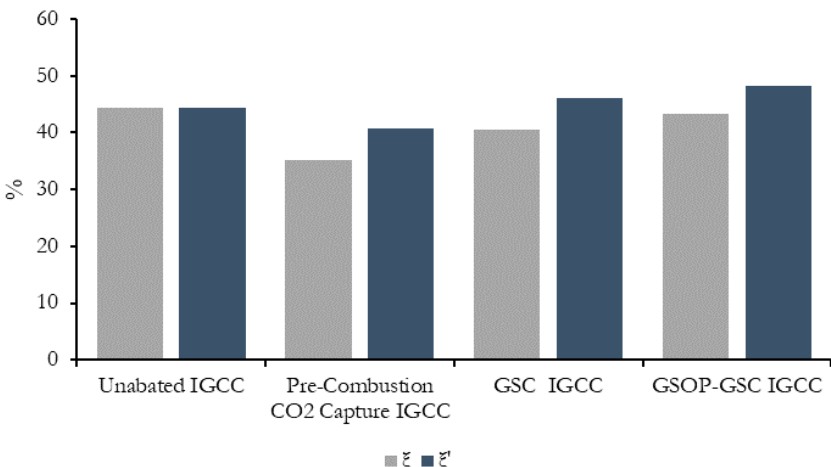

**Figure 10.** Exergy efficiencies of the different power plants.

Lastly, a comprehensive tool to visualize the different flows of exergy across the plant is their representation in a Sankey flow diagram, in which the different nodes of the plant are interconnected with exergy arrows with a size that is proportional to the magnitude of these streams. In this way, an insight of the proportion of exergy flows that are exchanged between the different subsystems is given. Since GS plants involving chemical looping are less known compared to the reference plants, in this work, such diagrams are shown in Figures 11 and 12 for the GSC IGCC and GSOP–GSC IGCC plants, respectively. A detailed breakdown per item of the different power plant models is presented in Table A6 in Appendix A.

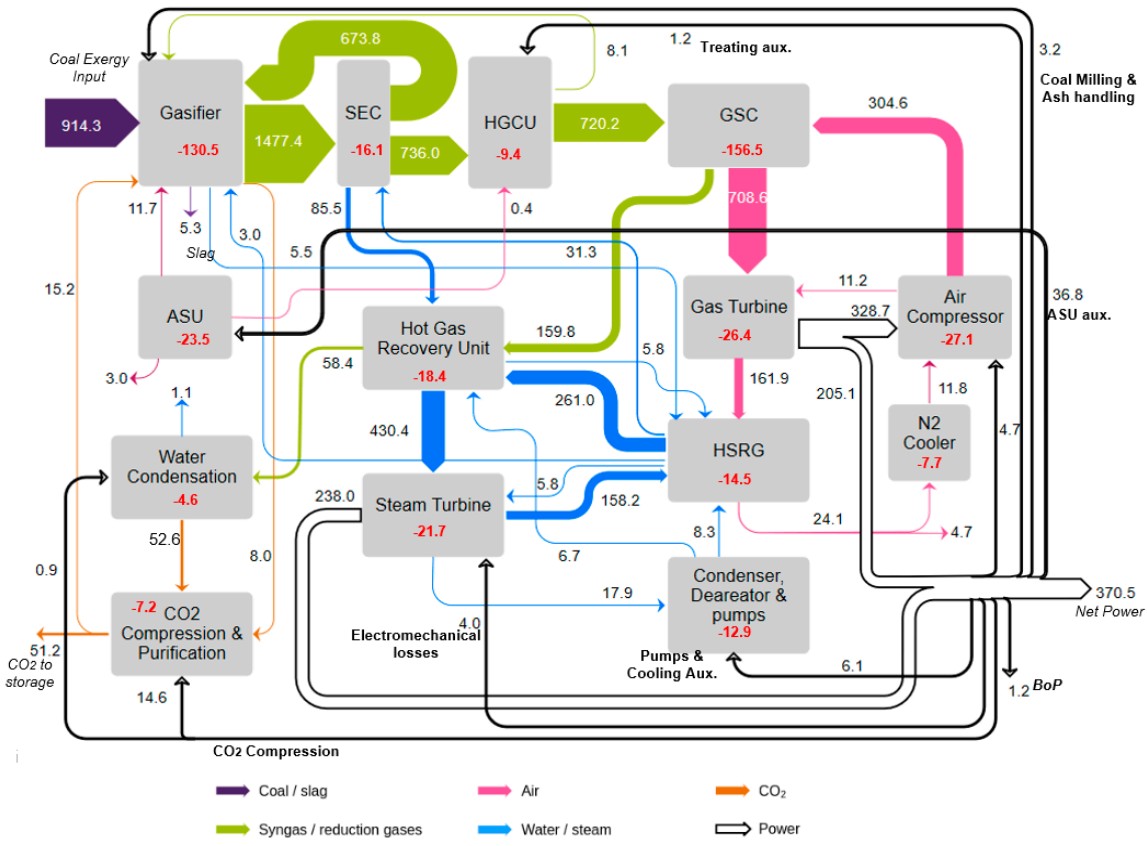

**Figure 11.** Sankey diagram of the exergy flow for the GSC IGCC power plant. Values are in MW. Red values represent exergy destruction.

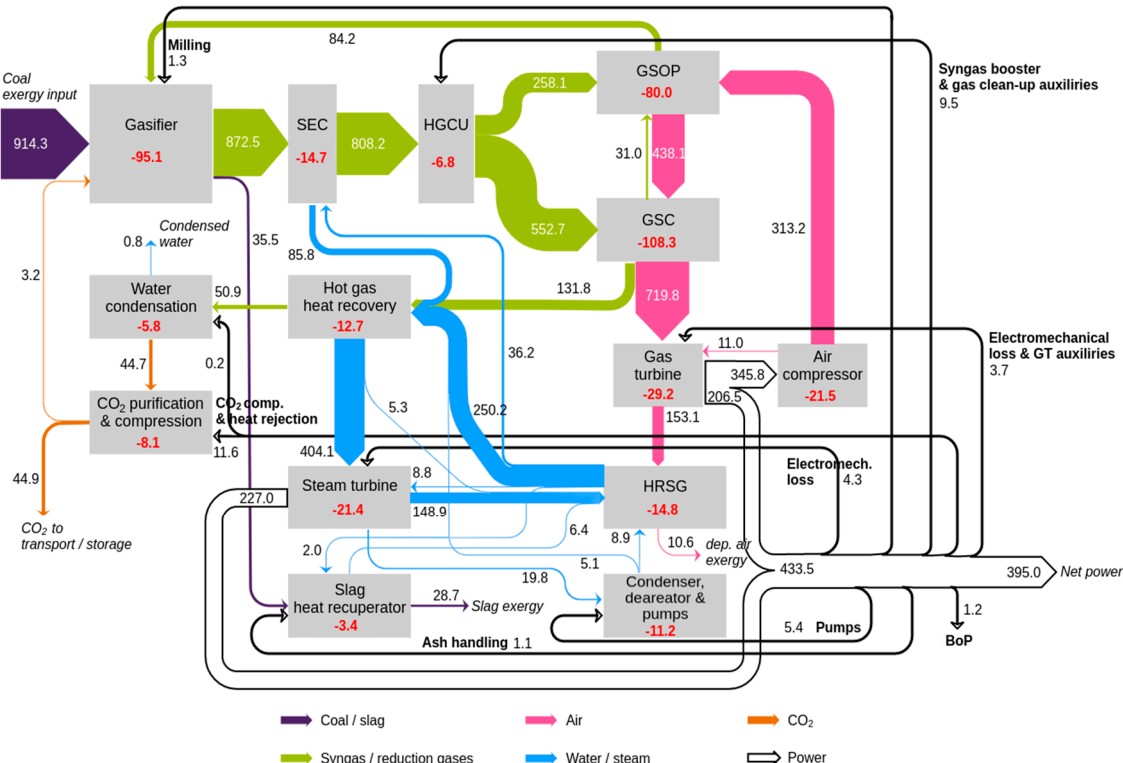

**Figure 12.** Sankey diagram of the exergy flow for the GSOP–GSC IGCC plant. Values are in MW. Red values represent the exergy destruction.

## 4. Summary and Conclusions

In this work, two IGCC power plant concepts with CCS based on gas switching chemical looping technology were investigated: an IGCC plant with a GSC cluster using NiO as the oxygen carrier, replacing the GT combustion chamber, and an IGCC plant where, additionally, the ASU was replaced by a GSOP cluster for oxygen production. The plants were consistently benchmarked with two reference IGCC technologies: a power plant with no $CO_2$ capture (or unabated IGCC) and a precombustion $CO_2$ capture IGCC power plant, with a WGS unit and $CO_2$ removal with Selexol. The GS chemical looping plants were modeled with HGCU for syngas desulfurization and contaminant removal, assuming that this treating technology will be available before GS technology deployment on a commercial scale.

The GSC IGCC plant shows a similar performance in terms of emissions and efficiency with respect to previous literature results [21]. The GSOP–GSC plant shows a thermal efficiency increase of approximately 1% point compared to the results presented in [10], because of the improved performance of the GSOP cluster operating in fluidization mode, but it led to a decrease of the capture ratio of approximately 10% points due to the increased outlet stream mixing of the clusters

Furthermore, a detailed exergy analysis of all the power plant configurations was presented, with the aid of a modeling auxiliary tool for the determination of physical and chemical exergy of the process streams. Additionally, since GS chemical looping plants are less known, a Sankey flow diagram of the exergy flows was presented for the GSC IGCC and GSOP–GSC IGCC plants. The exergy analysis methodology enabled the determination of the different sources of thermodynamic irreversibility across the power plants. The performance of the different plant sections was determined with the exergy utilization parameter $\lambda$ and the exergy loss contribution $\delta$, revealing that the largest exergy destruction takes place where the fuel is degraded to heat, i.e., the gasifier, combustion chamber, and GS clusters. The use of GS chemical looping technology improved the exergy utilization of the gas turbine section by avoidance of the combustion chamber; however, due to the lower TIT achieved, a larger portion of the total power output shifted to the bottoming steam cycle, and together with a

less efficient heat recovery of the reduction gases, it resulted in a lower exergy utilization and a higher exergy loss contribution of this section relative to the benchmark cases.

The exergy analysis revealed that the efficiency gains of the GSC IGCC plant relative to the precombustion $CO_2$ capture IGCC shown in earlier studies [21] can be to a great extent explained by the lower syngas cooling and treating exergy destruction values attained by means of inherent carbon capture (GSC) and HGCU. When a GSOP cluster is introduced, the foreseen low energy penalty suggested in [10] for the GSOP–GSC IGCC plant is corroborated through the exergy analysis, with a substantially lower exergy destruction taking place in the gasification island, due to the avoidance of the ASU and the lower operating temperature of the Winkler gasifier.

Based on these insights, several improvement pathways were suggested, which were primarily based on making better use of the exergy contained in the GSC reduction gases outlet. Optimization efforts focusing on the items where a large exergy destruction occurs have the most potential for improvement, therefore using this stream for coal pre-heating in the gasification island, are proposed and will be evaluated in future works. Although it is not the focus of the present study, it is noted that a parallel economic assessment is required to holistically determine the actual benefits of potential exergy gains. Regarding the overall exergy efficiency, when only the net power output of the plants is considered as the useful effect, the exergy efficiencies of the plants investigated were around 3% points lower than the respective thermal efficiency. When the exergy of the captured $CO_2$ was included as a useful effect, the exergy efficiency of the GS plants with CCS surpassed the unabated IGCC benchmark.

**Author Contributions:** C.A.d.P. developed the power plant models, performed the model analysis and wrote the article, elaborating most of the article figures. Á.J.Á. collaborated in the analysis of the power plants, performed a work of supervision and contributed to the elaboration of the exergy diagrams. J.H.C. elaborated the models of the transient reactors and provided input to the stationary power plant simulations, as well as text revision. S.C. performed a task of supervision and reviewing of the text, providing insights on modelling of the systems evaluated. S.A., as project coordinator, performed a task of supervision and revision of the text. All authors have read and agreed to the published version of the manuscript.

**Funding:** This research received financial support from the ERA-NET cofund, ACT GaSTech Project number 276321 co-funded by the European Commission under the Horizon 2020 program, and ACT Grant Agreement No. 691712. The partners collaborating in this article have received funding from MINECO, Spain (reference PCIN-2017-013) and the Research Council of Norway, Norway.

**Acknowledgments:** The authors would like to acknowledge Honeywell for the free Academic License of Unisim Design Suite R451, which enabled the integration of the GSOP and GSC models with power plant simulations.

**Conflicts of Interest:** The authors declare no conflict of interest.

## Abbreviations

| Nomenclature | |
|---|---|
| $\dot{E}$ | Total Exergy Flow (kW) |
| $\dot{I}$ | Exergy Destruction (kW) |
| $\dot{Q}$ | Heat Flow (kW) |
| $P$ | Pressure (bar) |
| $T$ | Temperature (K) |
| $\dot{W}$ | Power (kW) |
| $\dot{m}$ | Mass Flow (kg/s) |
| $h$ | Specific Enthalpy (kJ/kg) |
| $e$ | Exergy Flow (kJ/kg) |
| $s$ | Specific Entropy (kJ/kg K) |
| $\xi$ | Exergy Efficiency (%) |
| $\lambda$ | Exergy Utilization (%) |
| $\delta$ | Exergy Loss Contribution (%) |

**Subscripts & Superscripts**

| | |
|---|---|
| *i* | Stream |
| *j* | System |
| ch | Chemical |
| ph | Physical |
| 0 | Ambient Conditions |

**Acronyms**

| | |
|---|---|
| AGRU | Acid Gas Removal Unit |
| ASME | American Society of Mechanical Engineers |
| ASU | Air Separation Unit |
| BoP | Balance of the Plant |
| CCS | Carbon Capture and Storage |
| CGCU | Cold Gas Clean Up |
| CGE | Cold Gas Efficiency |
| CLC | Chemical Looping Combustion |
| CLOP | Chemical Looping Oxygen Production |
| CPU | Cryogenic Purification Unit |
| GSC | Gas Switching Combustion |
| GSOP | Gas Switching Oxygen Production |
| GT | Gas Turbine |
| HGCU | Hot Gas Clean Up |
| HP | High Pressure |
| HSRG | Heat Recovery Steam Generator |
| HX | Heat Exchanger |
| IGCC | Integrated Gasification Combined Cycle |
| IP | Intermediate Pressure |
| LHV | Lower Heating Value |
| LP | Low Pressure |
| LTHR | Low Temperature Heat Recovery |
| MITA | Minimum Temperature Approach |
| PB | Packed Bed |
| SEC | Syngas Effluent Cooler |
| SPECCA | Specific Primary Energy Cost of $CO_2$ Avoided |
| ST | Steam Turbine |
| TIT | Turbine Inlet Temperature |
| TOT | Turbine Outlet Temperature |
| WFGD | Wet Flue Gas Desulphurization |
| WGS | Water Gas Shift |

# Appendix A

**Table A1.** Gasification island assumptions.

| Winkler Gasifier | | |
|---|---|---|
| *Item* | *Value* | *Units* |
| Freeboard temperature | 900 | °C |
| %w. $CO_2$ for coal loading | 15 | % |
| % LHV $CH_4$ in syngas | 11.3 | % |
| Oxidizer overpressure | 50 | kPa |
| HP steam superheat | 450 | °C |
| Fixed carbon conversion | 95.5 | % |
| %w. Vented $CO_2$ in lock hoppers | 10 | % |
| Coal milling and handling | 40 | MJ/kg coal |
| Ash handling | 200 | MJ/kg ash |
| **Shell Gasifier** | | |
| *Item* | *Value* | *Units* |
| Moderator (steam) to dry coal ratio | 0.09 | kg/kg |
| Oxygen to dry coal ratio | 0.873 | kg/kg |
| Moisture in coal after drying | 2 | % |
| Syngas for coal drying %LHV | 0.9 | % |
| Fixed carbon conversion | 99.3 | % |
| Gasifier operating pressure | 44 | bar |
| Steam moderator pressure | 54 | bar |
| Heat loss as %LHV | 0.7 | % |
| Heat to membrane wall as %LHV | 2 | % |
| $CO_2$ HP/HHP pressure | 56/88 | bar |
| $CO_2$ temperature | 80 | °C |
| $CO_2$ to dry coal ratio | 0.83 | kg/kg |
| **Air Separation Unit** | | |
| *Item* | *Value* | *Units* |
| Main air compressor polytropic efficiency | 89 | % |
| Booster air compressor polytropic efficiency | 87 | % |
| Reboiler–condenser pinch | 1.5 | °C |
| Heat exchanger minimum approach temperature | 2 | °C |
| Process stream temperature after heat rejection | 25 | °C |
| Oxygen purity | 95 | % |
| Oxygen delivery pressure | 48 | bar |
| Oxygen pump efficiency | 80 | % |
| Exchanger pressure losses/side | 10 | kPa |
| Intercooler pressure loss | 10 | kPa |

**Table A2.** Syngas treating unit assumptions.

| HGCU | | |
|---|---|---|
| *Item* | *Value* | *Units* |
| Adsorption temperature | 400 | °C |
| Regeneration temperature | 750 | °C |
| Filter pressure drop | 5 | % |
| Auxiliary consumption | 5.34 | $MJe/kgH_2S$ |
| Compander polytropic efficiency | 80 | % |
| Syngas blower polytropic efficiency | 80 | % |
| $O_2$ mol fraction in regeneration stream | 2–5 | % |
| **CGCU** | | |
| *Item* | *Value* | *Units* |
| Absorption temperature | 30 | °C |
| Auxiliary consumption | 3 | $MJe/kg\ H_2S$ |
| LP steam requirement | 50 | $MJth/kg\ H_2S$ |
| Syngas blower polytropic efficiency | 80 | % |
| Selexol pump efficiency | 80 | % |
| % $H_2S$ to Claus unit | >25 | % |

**Table A3.** Power cycle assumptions.

| Gas Turbine | | |
|---|---|---|
| *Item* | *Value* | *Units* |
| GT compressor polytropic efficiency | 91.5 | % |
| GT turbine polytropic efficiency | 87 | % |
| GT pressure ratio | 18.1 | - |
| GT electromechanical efficiency | 98.6 | % |
| **Steam Cycle** | | |
| *Item* | *Value* | *Units* |
| ST LP stage isentropic efficiency | 88 | % |
| ST IP stage isentropic efficiency | 94 | % |
| ST HP stage isentropic efficiency | 92 | % |
| ST electromechanical efficiency | 98.1 | % |
| Pressure levels HP/IP/LP | 144/36/4 | bar |
| Auxiliaries for heat rejection | 0.008 | MJe/MJth |
| Pump isentropic efficiency | 80 | % |
| Live steam temperature | 565 | °C |
| **$CO_2$ Compression** | | |
| *Item* | *Value* | *Units* |
| $CO_2$ compressor stage isentropic efficiency | 80 | % |
| Process stream temperature after cooler | 25 | °C |
| Cold exchanger MITA | 2 | °C |

**Table A4.** Dry air composition.

| Component | Mole Fraction |
|---|---|
| $N_2$ | 0.7808 |
| $O_2$ | 0.2095 |
| Ar | 0.0093 |
| $CO_2$ | 0.0004 |
| $SO_2$ | 0.1 PPM |

**Table A5.** Douglas Premium coal properties.

| Ultimate Analysis | Mass Frac |
|---|---|
| C | 0.6652 |
| N | 0.0156 |
| H | 0.0378 |
| S | 0.0052 |
| O | 0.0546 |
| Cl | 0.00009 |
| Moisture | 0.08 |
| Ash | 0.1415 |
| Volatiles | 0.2291 |
| LHV (MJ/kg) | 25.17 |

Table A6. Exergy balance per item of the different power plant models. Total values are in MW.

| Model | | Unabated IGCC | | Pre-Combustion CO$_2$ Capture IGCC | | GSC IGCC | | GSOP-GSC IGCC | |
|---|---|---|---|---|---|---|---|---|---|
| *Block* | *Item* | *Total* | *%* | *Total* | *%* | *Total* | *%* | *Total* | *%* |
| | ASU | 17.9 | 2, 0 | 23.7 | 2.6 | 23.5 | 2.6 | 0.0 | 0.0 |
| | Air expander and cooling | 3.8 | 0.4 | 0.0 | 0.0 | 0.0 | 0.0 | 0.0 | 0.0 |
| **Gasification Island** | N$_2$ compression | 7.0 | 0.8 | 6.5 | 0.7 | 0.0 | 0.0 | 0.0 | 0.0 |
| | Gasifier (and quench) | 130.3 | 14.2 | 129.4 | 14.1 | 130.5 | 14.3 | 95.1 | 10.4 |
| | SEC (+Slag HX) | 15.9 | 1.7 | 16.2 | 1.8 | 16.1 | 1.8 | 18.1 | 2.0 |
| | HGCU (+Booster) | 0.0 | 0.0 | 0.0 | 0.0 | 9.4 | 1.0 | 6.8 | 0.7 |
| **Syngas Cooling and** | Scrubbing, LTHR and CGCU | 15.5 | 1.7 | 0.0 | 0.0 | 0.0 | 0.0 | 0.0 | 0.0 |
| **Treating** | Scrubber and WGS | 0.0 | 0.0 | 44.4 | 4.8 | 0.0 | 0.0 | 0.0 | 0.0 |
| | AGRU (Selexol) | 0.0 | 0.0 | 19.3 | 2.1 | 0.0 | 0.0 | 0.0 | 0.0 |
| | Saturator and fuel heating | 13.9 | 1.5 | 14.6 | 1.6 | 0.0 | 0.0 | 0.0 | 0.0 |
| **GSOP Cluster** | GSOP | 0.0 | 0.0 | 0.0 | 0.0 | 0.0 | 0.0 | 80.0 | 8.8 |
| **GSC cluster** | GSC | 0.0 | 0.0 | 0.0 | 0.0 | 156.5 | 17.1 | 108.3 | 11.8 |
| | Compressor | 18.1 | 2.0 | 16.2 | 1.8 | 27.1 | 3.0 | 21.5 | 2.4 |
| **Gas Turbine** | Combustion chamber | 165.9 | 18.1 | 163.0 | 17.8 | 0.0 | 0.0 | 0.0 | 0.0 |
| | Turbine | 33.2 | 3.6 | 31.4 | 3.4 | 26.4 | 2.9 | 29.2 | 3.2 |
| | HSRG | 19.3 | 2.1 | 13.2 | 1.4 | 14.5 | 1.6 | 14.8 | 1.6 |
| | Hot gases recovery unit | 0.0 | 0.0 | 0.0 | 0.0 | 18.4 | 2.0 | 12.7 | 1.4 |
| **Steam Cycle** | Condenser cooling and pumps | 9.2 | 1.0 | 7.8 | 0.9 | 12.9 | 1.4 | 11.2 | 1.2 |
| | N$_2$ cooler | 0.0 | 0.0 | 0.0 | 0.0 | 7.7 | 0.8 | 0.0 | 0.0 |
| | Steam turbine | 17.1 | 1.9 | 14.3 | 1.9 | 21.7 | 2.4 | 21.4 | 2.3 |
| **CO$_2$ Purification &** | Reduction gases condenser | 0.0 | 0.0 | 0.0 | 0.0 | 4.6 | 0.5 | 5.8 | 0.6 |
| **Compression** | CPU/CO$_2$ compression | 0.0 | 0.0 | 10.3 | 1.1 | 7.2 | 0.8 | 8.1 | 0.9 |
| | Slag | 5.3 | 0.6 | 5.3 | 0.6 | 5.3 | 0.6 | 28.7 | 3.1 |
| **Exergy Out** | CO$_2$ | 0.0 | 0.0 | 51.4 | 5.6 | 51.2 | 5.6 | 44.9 | 4.9 |
| | Stack gases | 28.7 | 3.1 | 20.1 | 2.2 | 4.7 | 0.5 | 10.6 | 1.2 |
| | Other (water, etc.) | 5.8 | 0.6 | 5.7 | 0.6 | 5.6 | 0.6 | 0.7 | 0.1 |
| **Other** | Misc. (BoP) | 1.3 | 0.1 | 1.3 | 0.1 | 1.3 | 0.1 | 1.3 | 0.1 |
| **Useful Effect** | Wnet | 406.9 | 44.5 | 322.7 | 35.2 | 370.5 | 40.5 | 395.2 | 43.2 |
| **Input** | Exergy in | 915.1 | 100.0 | 916.8 | 100.0 | 914.9 | 100.0 | 914.5 | 100.0 |

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
