# Peer review of "Exergy Analysis of Gas Switching Chemical Looping IGCC Plants"

_energies, doi:10.3390/en13030544_

Round 1

Reviewer 1 Report

The paper submitted to Energies is of interest for a particular readers only. It can be viewed that the development of the models has been performed for several months. However, the presentation of the exergy analysis falls short to what it was expected from a work like this. Hence the following points are given to be considered for possible publication:

At the onset, the authors make an introduction of the state-of- the-art technologies to achieve the climate change targets. That is good. But they do not provide a clear justification as to why their work should be read. What is the relevance of the current work?. What is the purpose of using an exergy method and not an energy one?. In reference to Figure 1, could you kindly explain the meaning of "Number of reduction stage lengths"?. In reference to Equation (1), why do the authors use "B" for total exergy and "e" for specific exergy. It is better to use B and b, or E and e. Please check on Kotas book for a better reference. Besides, what is the meaning of " B' and e' "?, in Equation (1). In Section 2.3, line 309, the authors comment that when a large amount of heat is rejected to the environment through a cooling device, it is assumed that all the exergy is destroyed in that unit... This assumption was made by the authors?, if so, why are the grounds to justify this?. If not, where was it drawn from?  In reference to description of Figure 7, it is not clear if the figure represents exergy losses or exergy consumption?, it is confusing. It is not clear the idea of providing both exergy breakdown and % exergy breakdown?, they both are the same information. It should be better to provide the another parameter of exergy performance such as, for example, the exergy rational efficiency. In reference to section of conclusions, the authors mentioned that they benchmarked the technologies from a thermal and an environmental perspective. Where are the environmental results?, these are not provided in the paper, if any. It is not always a rule to focus on the items where considerable exergy destruction is caused. Or how do the authors justify their comment?.

Reviewer 2 Report

Review

“Exergy Analysis of Gas Switching Chemical Looping 2 IGCC plants”

Carlos Arnaiz de Pozo, Angel Jiménez Álvaro , Jan Hendrik Cloete, Schalk Cloete and Shahriar Amini

Main idea

Main idea is to analyse IGCCsystems that comprise CCL and or CLOP is novelty in literature and worthy of looking at. Stated aim “the exergy analysis methodology is employed to evaluate the performance of the novel GSC IGCC and the GSOP-GSC IGCC power plants and benchmark them against an Unabated IGCC plant and an IGCC plant with pre-combustion CO2 capture.”

2.1 Reactor simulations

In Figure 1 you have 178 s for a) and 253.7 s for b). What is the reason for the added accuracy in b)?

3.3.1 Plant Power Breakdown

To properly evaluate one needs the air & flue gas flows. Please add these to e.g. bottom of the Table 1.

Conclusions

Clarify what you mean by different IGCC plants. In the abstract acronyms CCL and CLOP are mentioned. The same is done in the Introduction. But these acronyms do not appear anywhere in the Conclusions. Please revise.

Errata

The authors are suggested to carefully go though all subscripts as probably not all are listed below

33               I Exergy Destruction (kJ)  see tabs

Subscripts & Superscripts  missing e and th

295             subscripts “B’j”

304             subscripts “ei,j”

313             space “e_pand” subscripts “e_p”, “e_c”, “e_Ch”, “e_Ph”

315             subscripts “Pi” “Ti” “P0” and “T0”,

455             “Treating auxiliaries slightly increase for this plant compared” Increase what?

507             Condesner

519             ”where” were

539             “This section is not mandatory, but can be added to the manuscript if the discussion is unusually long or complex.”  ????

566             2. Ishida, M., Zheng, D., Akehata, T., order of surname and first letter, and

567             system by graphic exergy analysis. " Energy, pp. 12, 147-154. remove . 

580             “CO 2 capture”

616             “Practice Guidelines For Assessment Of Co2 Capture Technologies,". caps, CO2

623             “CO 2 ”

631             '"Chapter 4 - CYCLE EFFICIENCY WITH TURBINE COOLING (COOLING FLOW RATES SPECIFIED),"  “ “” “, caps

Reviewer 3 Report

In the paper "Exergy Analysis of Gas Switching Chemical Looping IGCC plants", the authors performed exergy analyses of a GSC IGCC plant and a GSOP–GSC IGCC plant. Those analyses are benchmarked against an Unabated IGCC and a Pre-combustion CO2 capture IGCC plant.
In the theoretical part of the paper, the authors described gasification step to generate a clean syngas fuel and different solutions to make them more efficient.
In the practical part of the paper a GSC IGCC and GSOP-GSC IGCC plant are modelled. Simulations of thermal analysis, power breakdown and emission performance for the different power plants were prepared. Considering the fact that heat and work are not fully interchangeable, in the paper rational efficiency has been introduced in the prepared exergy analysis.
Because of the environment, the generation of clean energy is very important nowadays, so exergy analysis prepared in this article are interesting and originality.
The structure and content of the paper is clear.

Reviewer 4 Report

The subject described in the paper is very important and exteremely interesting due to nowadays increasing requirements for power production systems. First sentence od reviewed work: Integrated Gasification Combined Cycles (IGCC) are promising power production systems from solid fuels due to their high efficiency and good environmental performance is obvious true and developing of these systems should be in high concern of modern science and industry. Proposed consideration of comparing systems seems to be very useful and conducted in a good scientific manner. Proposed approach maybe is not of highest originality but its practical and very needful for developing power production systems therefore publishing of reviewed paper is highly recommended. From very minor suggestions: figure 6 could be improved - low quality image makes a little dificulty in reading.

Round 2

Reviewer 1 Report

Dear authors, the changes you have done to the paper are acceptable and recommend the paper to be accepted.